# Ocular conjunctival inoculation of SARS-CoV-2 can cause mild COVID-19 in rhesus macaques

Wei Deng[1,3], Linlin Bao[1,3], Hong Gao[1,3], Zhiguang Xiang[1,3], Yajin Qu[1,3], Zhiqi Song[1,3], Shuran Gong[1], Jiayi Liu[2], Jiangning Liu[1], Pin Yu[1], Feifei Qi[1], Yanfeng Xu[1], Fengli Li[1], Chong Xiao[1], Qi Lv[1], Jing Xue[1], Qiang Wei[1], Mingya Liu[1], Guanpeng Wang[1], Shunyi Wang[1], Haisheng Yu[1], Ting Chen[1], Xing Liu[1], Wenjie Zhao[1], Yunlin Han[1] & Chuan Qin [1✉]

Severe acute respiratory syndrome coronavirus 2 (SARS-CoV-2) is highly transmitted through the respiratory route, but potential extra-respiratory routes of SARS-CoV-2 transmission remain uncertain. Here we inoculated five rhesus macaques with $1 \times 10^6$ TCID$_{50}$ of SARS-CoV-2 conjunctively (CJ), intratracheally (IT), and intragastrically (IG). Nasal and throat swabs collected from CJ and IT had detectable viral RNA at 1–7 days post-inoculation (dpi). Viral RNA was detected in anal swabs from only the IT group at 1–7 dpi. Viral RNA was undetectable in tested swabs and tissues after intragastric inoculation. The CJ infected animal had a higher viral load in the nasolacrimal system than the IT infected animal but also showed mild interstitial pneumonia, suggesting distinct virus distributions. This study shows that infection via the conjunctival route is possible in non-human primates; further studies are necessary to compare the relative risk and pathogenesis of infection through these different routes in more detail.

[1] NHC Key Laboratory of Human Disease Comparative Medicine, Beijing Key Laboratory for Animal Models of Emerging and Remerging Infectious Diseases, Institute of Laboratory Animal Science, Chinese Academy of Medical Sciences and Comparative Medicine Center, Peking Union Medical College, Beijing, China. [2] Department of Radiology, Bejing Anzhen Hospital, Capital Medical University, Beijing, China. [3] These authors contributed equally: Wei Deng, Linlin Bao, Hong Gao, Zhiguang Xiang, Yajin Qu, Zhiqi Song. ✉email: qinchuan@pumc.edu.cn

Coronavirus disease 2019 (COVID-19) is highly infectious and transmitted mainly through human-to-human transmission via respiratory droplets when in direct or close contact with patients with severe acute respiratory coronavirus 2 (SARS-CoV-2)[1]. Other potential SARS-CoV-2 transmission routes remain to be further researched. In some clinical cases, samples of tears and conjunctival secretions from both SARS-CoV[2] and SARS-CoV-2 patients with conjunctivitis[3] displayed detectable viral RNA. A previous report demonstrated that among 38 patients with clinically confirmed COVID-19, a total of 12 patients had ocular manifestations consistent with conjunctivitis, and conjunctival and nasopharyngeal specimens from 2 of these 12 patients were positive for SARS-CoV-2 (ref. [4]). Another study reported the case of a clinician infected with SARS-CoV-2 while working with patients under all safeguards except eye protection[3]. In contrast, no SARS-CoV-2 could be detected by reverse transcription PCR (RT-PCR) in 114 conjunctival swab samples from patients with COVID-19 pneumonia[5]. On the other hand, a set of atypical patients had gastrointestinal symptoms at the onset of infection but were without fever or other respiratory symptoms[6]. Based on single-cell transcriptome-bioinformatics analysis, researchers have revealed that the digestive system is a potential route of SARS-CoV-2 infection[7]. Another experiment performed with human small intestinal organoids demonstrated that enterocytes were readily infected by SARS-CoV-2 and produced infectious viral particles[8]. Potential extra-respiratory portals through which SARS-CoV-2 enters the host need to be further researched by laboratory confirmation to provide valuable data for oversight and prevention for healthcare workers and the population.

Here, we report that infection via the conjunctival route is possible in non-human primates and that respiratory droplets are the main route of SARS-CoV-2 infection. The conjunctivally (CJ)-infected animal has a higher viral load in the nasolacrimal system than the intratracheally (IT)-infected animal but mild interstitial pneumonia, suggesting distinct viral distributions. Viral RNA is undetectable in tested swabs and tissues after intragastric inoculation.

## Results

**SARS-CoV-2 replication and distributions.** Five rhesus macaques between the ages of 3 and 5 years were inoculated with $1 \times 10^6$ 50% tissue-culture infectious doses (TCID$_{50}$) of SARS-CoV-2 via three routes: ocular conjunctival inoculation (CJ-1 and CJ-2), intragastric inoculation (IG-1 and IG-2), and intratracheal inoculation (IT-1). IT-1 was used to compare the distribution and pathogenesis of the virus after entering the host via different routes (Fig. 1). The low number of non-human primates and the single tested dose is a limitation of the study and that therefore the risk of ocular exposure in comparison to other transmission routes of infection cannot be assessed.

We observed the macaques daily for clinical signs. There was no significant change in body weight (Fig. 2a) or temperature (Fig. 2b) in any of the inoculated macaques. Routine specimens, including nasal and throat swabs, were collected at 0, 1, 3, 5, and 7 days post-inoculation (dpi). Additionally, to explore the potential excretory routes of SARS-CoV-2 in the host, conjunctival and anal swabs were also gathered. Specifically, a continuous detectable viral load was observed in nasal and throat swabs collected from CJ- and IT-inoculated animals from 1 to 7 dpi. In contrast, the virus was not detected in the swabs collected from the IG-inoculated macaques. Notably, only with infection via the conjunctival route could the viral load be detected in conjunctival swabs (average, approximately 4.33 log$_{10}$ RNA copies/mL) collected on 1 dpi, after which the viral load became

undetectable. Furthermore, only with infection via the intratracheal route could the viral load be continuously detected in anal swabs collected from 1 to 7 dpi, with the viral load peaking at approximately 6.76 log$_{10}$ RNA copies/mL on 5 dpi, revealing the distinct excretory pathways of the host after inoculation via different routes (Fig. 2c).

To determine the distribution of virus and evaluate histological lesions, CJ-1, IT-1, and IG-1 were euthanized and necropsied at 7 dpi. For CJ-1, the viral load was primarily distributed in the nasolacrimal and ocular system (1.01–3.93 log$_{10}$ RNA copies/mL), including the lacrimal gland, optic nerve, and conjunctiva; nose (1.03–6.63 log$_{10}$ RNA copies/mL), including the nasal mucosa, nasal turbinate, and nostril; pharynx (4.57–5.60 log$_{10}$ RNA copies/mL), including the epiglottis and soft palate; oral cavity, including the cheek pouch and parotid gland; and other tissues, including the lower left lobe of the lung (4.6 log$_{10}$ RNA copies/mL), tonsil, inguinal and pararectal lymph nodes, stomach, duodenum, ileum, and caecum (Fig. 1d). In contrast, for IT-1, the distribution of the virus was somewhat different because viral replication was high in the different lobes of the lung (4.22–7.81 log$_{10}$ RNA copies/mL), and the viral load was also widely detected in the nasal septum (4.69 log$_{10}$ RNA copies/mL); trachea (6.61 log$_{10}$ RNA copies/mL); lymph organs and tissues (3.39–5.17 log$_{10}$ RNA copies/mL), including the mandibular lymph nodes, tonsils, and pulmonary lymph nodes; and some segments of the alimentary tract (1.11–4.93 log$_{10}$ RNA copies/mL), including the duodenum, ileum, colon, and caecum (Fig. 2d). There was no detectable viral load in the tissues collected from IG-1 (Supplementary Table 1). Furthermore, to evaluate the infectious virus titre in the lung at 7 dpi, virus collected from different lobes of the lungs of CJ-1 and IT-1 was inoculated onto Vero E6 cells for virus isolation. For CJ-1, SARS-CoV-2 was isolated from only the left lower lobe of the lung (2.67 log$_{10}$ TCID$_{50}$/mL). For IT-1, the virus was isolated from more lobes of the lung (2.00–5.67 log$_{10}$ TCID$_{50}$/mL), most of which showed higher virus titres than those in CJ-1. These were the left lower lobe, right lower lobe, left middle lobe, right middle lobe, right accessory lobe, left upper lobe, and right upper lobe of the lung (Fig. 2e). The distinct distributions of the virus inoculated by different routes were consistent with the anatomical structure, suggesting that the distribution of SARS-CoV-2 in the host is associated with the infection route. Furthermore, the virus was detectable in different segments of the alimentary tract and some lymphatic tissues in animals inoculated via both routes, revealing that these tissues may play important roles in the spread of the virus within the host[8]. Furthermore, specific IgG antibody against SARS-CoV-2 was detectable in CJ-2 at 14 and 21 dpi but not before infection, demonstrating that the animal had been infected with SARS-CoV-2 (Fig. 2f).

**Chest X-ray and histopathological changes.** Meanwhile, to observe the progressive pulmonary infiltration characteristic of SARS-CoV-2-related pneumonia, chest radiographs of the inoculated animals were recorded every other day after inoculation. Beginning at 3 dpi, abnormalities with varying degrees of severity appeared in the lungs. Specifically, for the CJ-inoculated animal, unlike the results collected before infection (day 0), the left upper lobe of the lung was presented opaque glass sign on 3 dpi, which then developed in the bilateral upper lobes on 5 dpi. Obscure lung markings and an opaque glass in the bilateral lobes of the lung were observed at 7 dpi. In contrast, the IT-inoculated animals developed relatively severe progressive pulmonary infiltration from 3 to 7 dpi. The right upper lobe of the lung exhibited an increased density and was obscure at 3 dpi. The right lower lobe of the lung presented obscure lung markings and lamellar

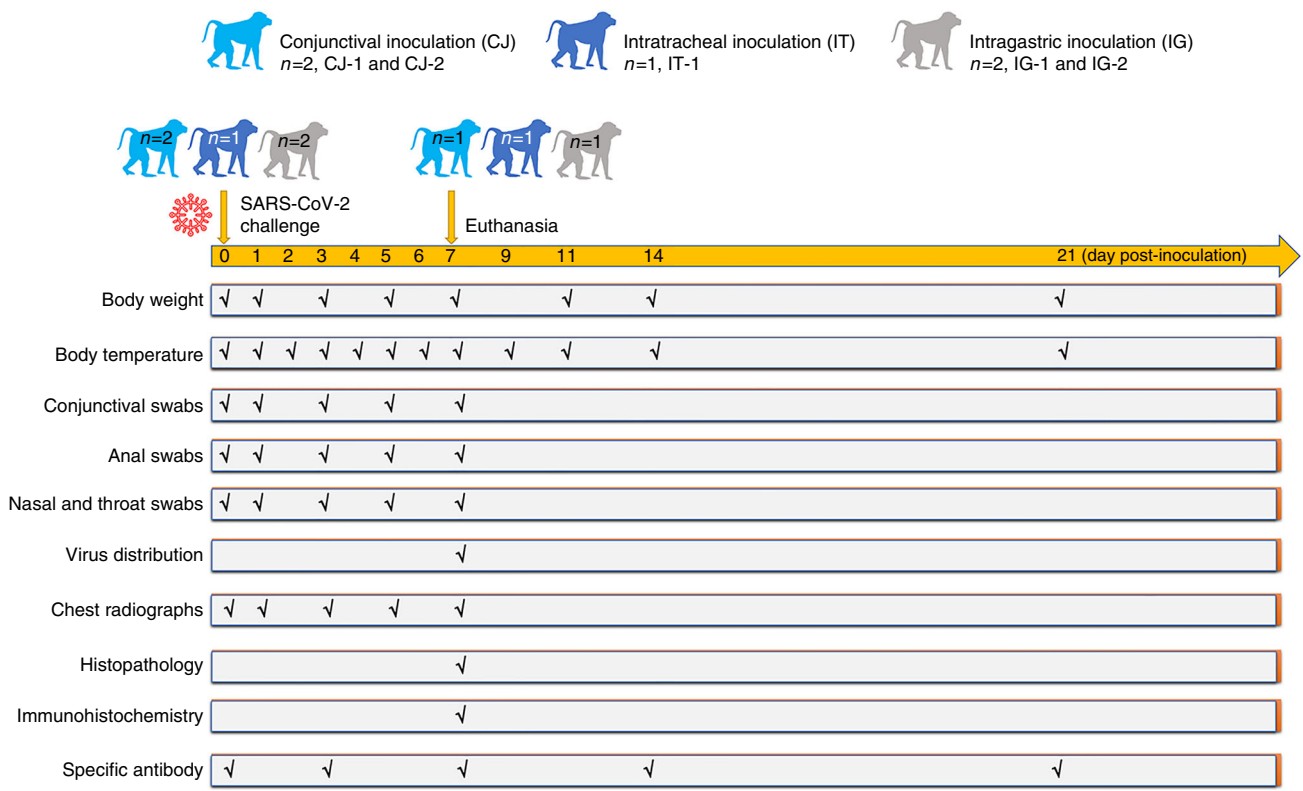

**Fig. 1 Graphical outline of the experimental design and sample collection.** Five male rhesus macaques (*Macaca mulatta*) were inoculated with $10^6$ TCID$_{50}$/mL SARS-CoV-2. Two rhesus macaques were inoculated via the ocular conjunctival route and named CJ-1 and CJ-2, one was inoculated via the intratracheal route and named IT-1, and two were inoculated via the intragastric route in sequence and named IG-1 and IG-2. The macaques were observed for clinical signs (body weight and temperature were tested as shown). On 0, 1, 3, 5, and 7 dpi, conjunctival, nasal, throat, and anal swabs were collected. CJ-1, IT-1, and IG-1 were euthanized and necropsied at 7 dpi. Tissues were collected to analyse the viral loads and titres. Serum was collected from all animals at 0 and 7 dpi and from CJ-2 and IG-2 at 14 and 21 dpi to examine the specific IgG antibodies against SARS-CoV-2 antigens.

ground-glass opacities at 5 dpi. Increased radiographic changes showing patchy lesions in the right upper lobe of the lung, obscure lung markings, and marked ground-glass opacities with a blurred right diaphragm were observed in the bilateral lobes of the lung at 7 dpi (Fig. 3a). Consistent with these radiographic alterations, upon microscopy examination, local lesions in the lungs of CJ-1 displayed mild interstitial pneumonia characterized by a thickened alveolar interstitium; the infiltration of inflammatory cells, primarily lymphocytes and monocytes; and a small amount of exudation in the alveolar cavities. IT-1 developed moderate and diffuse interstitial pneumonia characterized by a more thickened alveolar interstitium and more serious inflammation and exudation (Fig. 3b)[9]. Virus antigen was further confirmed by immunohistochemistry (IHC) staining with SARS-CoV-2-specific antibody. In the damaged lobes of the lungs of both CJ-1 and IT-1, SARS-CoV-2 was predominantly observed in the alveolar epithelium and exfoliated degenerative cellular debris in the alveolar cavities (Fig. 3b). Notably, the IHC results were highly consistent with the viral load detection data at 7 dpi collected from both CJ-1 and IT-1. Specifically, viral antigen was scattered in several cells in the nasolacrimal system via conjunctival inoculation but more prominent in the trachea following intratracheal inoculation (Fig. 4). Moreover, viral antigen was clearly observed in the lamina propria of the alimentary tract, suggesting that the virus can spread from the initial point of entry to gut-associated lymphoid tissue (Supplementary Fig. 1) and detected at low levels in the kidney, myocardium, and liver of IT-1 but was undetectable in CJ-1 (Supplementary Fig. 2). No substantial histopathological changes were observed in IG-1 (Supplementary Fig. 3). Briefly, the CJ-inoculated animal showed

evidence of relatively mild and local interstitial pneumonia, unlike the IT-inoculated macaque, which demonstrated moderate, diffuse lesions in the lung accompanied by the increased infiltration of inflammatory cells and accumulation of exudation in the alveolar cavities.

These data demonstrate that macaques can be infected with SARS-CoV-2 via the conjunctival and intratracheal routes but not the intragastric route. Compared to the intratracheal route, with the conjunctival route, the viral load in the nasolacrimal system was higher, and the lesions in the lung were milder and local.

## Discussion

We inoculated rhesus monkeys via the conjunctival, intragastric, or intratracheal route, avoiding co-inoculation via multiple routes, to confirm the exact pathway of inoculation. These results suggest that the conjunctiva is a potential portal of viral transmission. As shown by our results, viral load was detectable in several tissues associated with the nasolacrimal system, especially the conjunctiva, lacrimal gland, nasal cavity, and throat, which form an anatomical bridge between the ocular and respiratory tissues. In particular, the lacrimal duct functions as a conduit to collect tear fluid from the ocular surface and transport it to the nasal inferior meatus, which is convenient for the drainage of virus from ocular tissues to respiratory tract tissues. In fact, a previous report demonstrated that although virus-containing fluid could be taken up through the conjunctiva, sclera, or cornea, the majority of a liquid, including tears and secretions, is drained into the nasopharyngeal space or swallowed; the lacrimal duct epithelia can also contribute to the absorption of tear fluid. Our

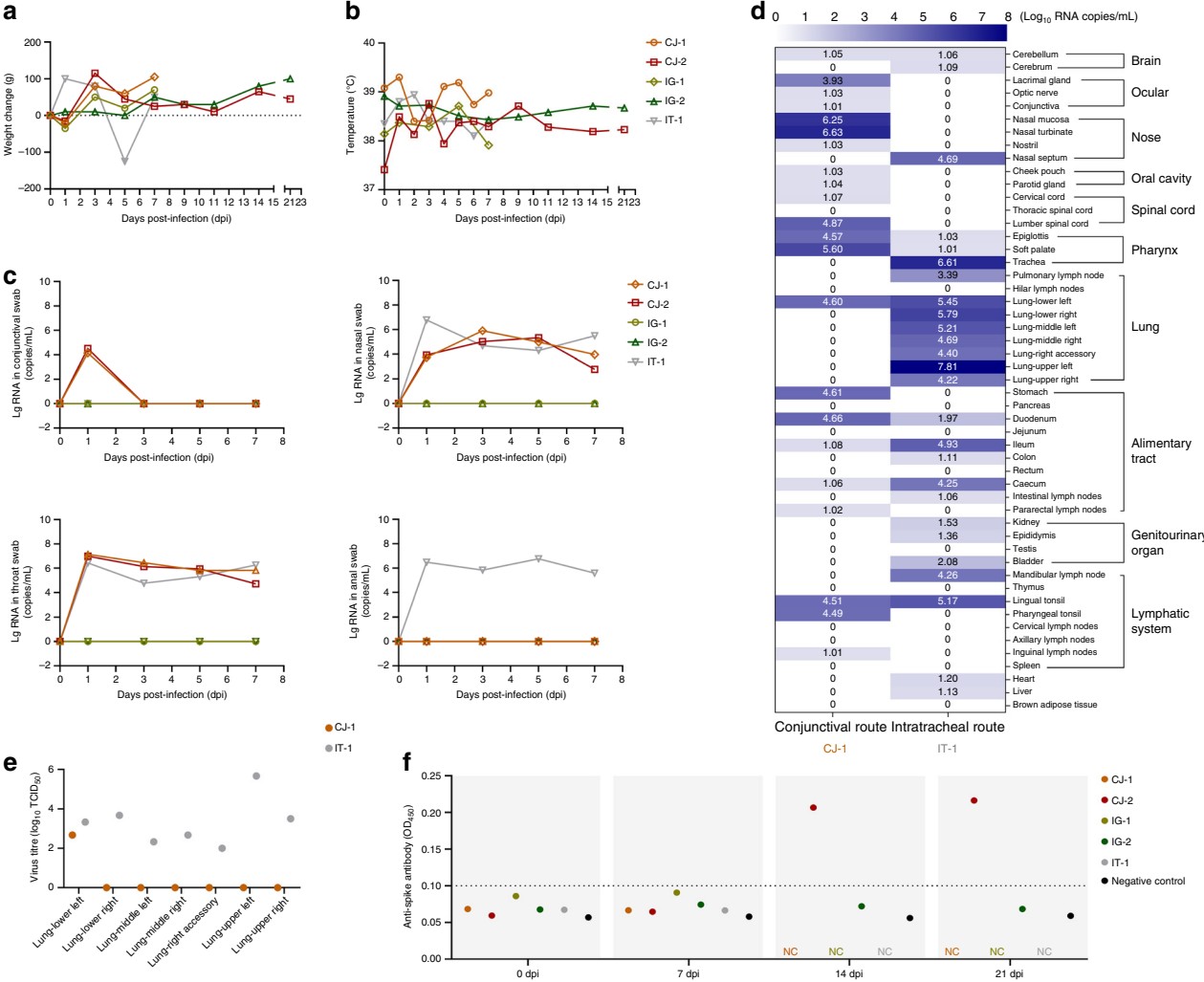

**Fig. 2 Clinical features, virus distributions, virus titres, and antibody detection.** Clinical signs, including body weight (**a**) and temperature (**b**), were observed. The viral loads of the conjunctival, nasal, throat, and anal swab specimens (**c**) from the five inoculated macaques collected at 0, 1, 3, 5, and 7 dpi were determined. The viral distribution in the majority of organs and tissues (**d**) and the virus titre in different lobes of the lungs (**e**) of CJ-1 and IT-1 collected on 7 dpi were tested and compared. The darker the blue colour is, the higher the viral load is. ELISA was used to detect specific IgG antibody against SARS-CoV-2 at 0, 7, 14, and 21 dpi (**f**). NC not collected.

results are highly consistent with the anatomical features that allow a virus to enter the host via the conjunctival route. However, further studies with more animals and different viral doses are needed to evaluate the relative risk of ocular exposure compared to other routes. On the other hand, previous research demonstrated that SARS-CoV was undetectable in cynomolgus monkeys after intragastric inoculation, which is consistent with our results. At present, no evidence has proven that SARS-CoV-2 can be transmitted via the faecal–oral route[10], although SARS-CoV-2 viral RNA was detectable in faecal samples from patients[11] and anal swabs from infected macaques for a prolonged time. Additionally, although a single animal was IT inoculated here, the viral load and pathological changes in the lungs of the IT-infected macaque in this study were in the range and consistent with the results of our previous studies in which rhesus macaques were IT infected with SARS-CoV-2. These previous data demonstrated that after euthanization and necropsy at 7 dpi, more than 4.0 $\log_{10}$ RNA copies/mL SARS-CoV-2 was detected in the lungs of IT-inoculated animals. Microscopically, the observed lesions were mainly in the lungs, where moderate interstitial pneumonia typically presents[9,12,13].

Respiratory viruses can stimulate ocular complications in infected patients, leading to respiratory infection[14]. The fact that exposure of the mucous membranes and a lack of eye protection increased the risk of SARS-CoV[2] or SARS-CoV-2 (ref. [15]) transmission suggests that increased awareness regarding the need for healthcare professionals in close contact with patients and individuals in crowded places to utilize eye protection is required.

## Methods

**Ethics statement**. The animal biosafety level 3 (ABSL3) facility at the Institute of Laboratory Animal Science was used to complete all the experiments with rhesus macaques. All research was performed in compliance with the Animal Welfare Act and other regulations relating to animals and experiments. The Institutional Animal Care and Use Committee of the Institute of Laboratory Animal Science, Peking Union Medical College, reviewed and authorized all the protocols in this research, including research done in animals (BLL20009).

**Cells and viruses**. A stock of the SARS-CoV-2 virus (SARS-CoV-2/WH-09/human/2020/CHN, accession No. MT093631.2)[16] was used in this study and cultivated in Vero E6 cells maintained in Dulbecco's modified Eagle's medium (DMEM; Invitrogen, Carlsbad, USA) supplemented with 10% foetal bovine serum (FBS) and incubated at 37 °C and 5% $CO_2$ (ref. [16]).

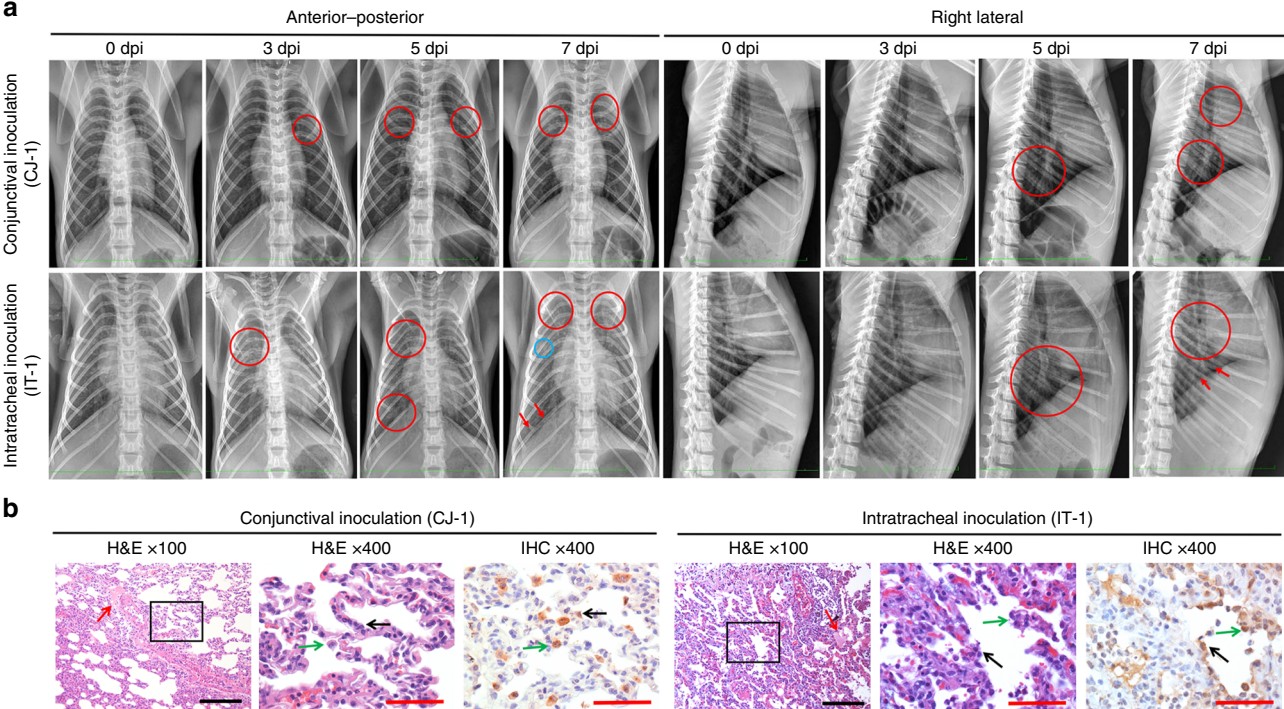

**Fig. 3 Comparison of lesions in the lungs of CJ-1 and IT-1.** Anterior–posterior and right lateral chest radiographs (**a**) were collected from rhesus macaques prior to SARS-CoV-2 inoculation (day 0) and at 3, 5, and 7 dpi. Areas of interstitial infiltration, indicative of pneumonia, are highlighted (red circle); obscure costophrenic angles (red arrows) and patchy lesions (blue circle) are also shown. The histopathological and immunohistochemical observations in the lungs are shown (**b**). Both macaques exhibited interstitial pneumonia with a thickened alveolar septum; the infiltration of inflammatory cells, mainly lymphocytes and macrophages; and some amount of exudation (red arrows) in the alveolar cavities at 7 dpi. Infection via the conjunctival route caused relatively mild pneumonia. Sequential sections were stained by H&E and subjected to IHC. Viral antigens were observed primarily in the alveolar epithelia (black arrows), along with detached degenerative cellular debris (green arrows). H&E-stained sections are shown at ×100 magnification and magnified at ×400 magnification (black frame). The IHC image shows the same field outlined in black at ×400 magnification. Black scale bar = 100 μm, red scale bar = 50 μm. Data (**b**) are representative of three independent experiments.

**RNA extraction and quantitative RT-PCR**. Total RNA was extracted from all of the collected organs[13,17] using the RNeasy Mini Kit (Qiagen, Hilden, Germany), and RT was performed with the PrimerScript RT Reagent Kit (TaKaRa, Japan) following the manufacturer's instructions. Quantitative real-time RT-PCR was performed using the PowerUp SYBR Green Master Mix Kit (Applied Biosystems, USA), and samples were processed in duplicate with the following cycling protocol: 50 °C for 2 min; 95 °C for 2 min; 40 cycles of 95 °C for 15 s, 60 °C for 30 s, and 95 °C for 15 s; 60 °C for 1 min; and 95 °C for 45 s. The primer sequences used for RT-PCR targeted the envelope (E) gene of SARS-CoV-2 and are as follows: forward: 5′-TCGTTTCGGAAGAGACAGGT-3′, reverse: 5′-GCGCAGTAAGGATG GCTAGT-3′. The PCR products were verified by sequencing with the dideoxy method on an ABI 3730 DNA sequencer (Applied Biosystems, CA, USA). During the sequencing process, amplification was performed with specific primers. The sequences of the primers used for this process are available upon request. The obtained sequencing reads were linked with DNAMAN, and the results were compared with the MEGALIGN module in the DNAStar software package.

**Animal experiments**. Five male rhesus macaques (*Macaca mulatta*) between the ages of 3 and 5 years were used in this research. All of them were negative for tuberculosis and simian immunodeficiency virus. They were inoculated with 1 × 10⁶ TCID₅₀ of SARS-CoV-2 via three routes. Two male rhesus macaques (named CJ-1 and CJ-2) were inoculated with 10⁶ TCID₅₀/mL SARS-CoV-2 via the ocular conjunctival route, one (named IT-1) was inoculated via the intratracheal route, and two (named IG-1 and IG-2) were inoculated via the intragastric route in sequence. Prior to sample collection, all animals were anaesthetised with ketamine hydrochloride (10 mg/kg). On 0, 1, 3, 5, and 7 dpi, conjunctival, nasal, throat, and anal swabs were collected and incubated in 1 mL of DMEM containing 50 μg/mL streptomycin and 50 U/mL penicillin. CJ-1, IT-1, and IG-1 were euthanized and necropsied at 7 dpi. Tissue samples were collected from the following organs to detect the viral load to analyse the distribution of the virus: the conjunctiva, lacrimal gland, optic nerve, cerebellum, cerebrum, different segments of the spinal cord, nostril, nasal turbinate, nasal mucosa, nasal septum, soft palate, cheek pouch, parotid gland, epiglottis, lingual tonsil, pharyngeal tonsil, different lobes of the lung, trachea, different lymph nodes, heart, liver, spleen, pancreas, different

segments of the alimentary tract, kidney, bladder, testis, and brown adipose tissue. Serum was collected from all animals at 0 and 7 dpi and from CJ-2 and IG-2 at 14 and 21 dpi for serological detection to examine specific IgG antibodies against the SARS-CoV-2 antigen.

**Preparation of homogenate supernatant**. An electric homogenizer was applied to prepare tissue homogenates by incubation in 1 mL of DMEM for 2.5 min. The homogenates were centrifuged at 825 × *g* at 4 °C for 10 min. The supernatants were harvested and stored at −80 °C to assess the v titre. To evaluate the infectious virus titre, homogenates were prepared from different lobes of the lungs from CJ-1 and IT-1 for virus titration analysis by endpoint titration in Vero E6 cells. The virus titres of the supernatants were determined using a standard TCID₅₀ assay[17].

**TCID₅₀ assay**. The TCID₅₀ assay was performed as following[18], briefly, to measure the SARS-CoV-2 titres, 10-fold serial dilutions of the virus were used to inoculate Vero cell monolayers in DMEM containing 2% FBS, which were incubated at 37 °C for 4 days. Then, the cytopathic effect was observed, and the TCID₅₀ values were calculated by the Reed and Muench method[19].

**Enzyme-linked immunosorbent assay (ELISA) to detect antibodies**. Serum was collected from each experimental animal at 0, 7, 14, and 21 dpi to detect SARS-CoV-2 antibodies through ELISA. Ninety-six-well plates that had been coated with 0.1 μg of the spike protein of SARS-CoV-2 (Sino Biological, 40591-V08H) and incubated at 4 °C overnight were blocked with 2% BSA/PBST at room temperature for 1 h. Serum samples were diluted 1:100, added to different wells, and maintained at 37 °C for 30 min, followed by incubation with goat anti-monkey antibody labelled with horseradish peroxidase (Abcam, ab112767, 1:10,000) at room temperature for 30 min. The absorbance at 450 nm was then determined.

**Haematoxylin and eosin (H&E) staining**. A ten percent buffered formalin solution was used to fix all the collected organs, and paraffin sections (3–4 μm in thickness) were prepared according to routine practice. All the tissue sections were

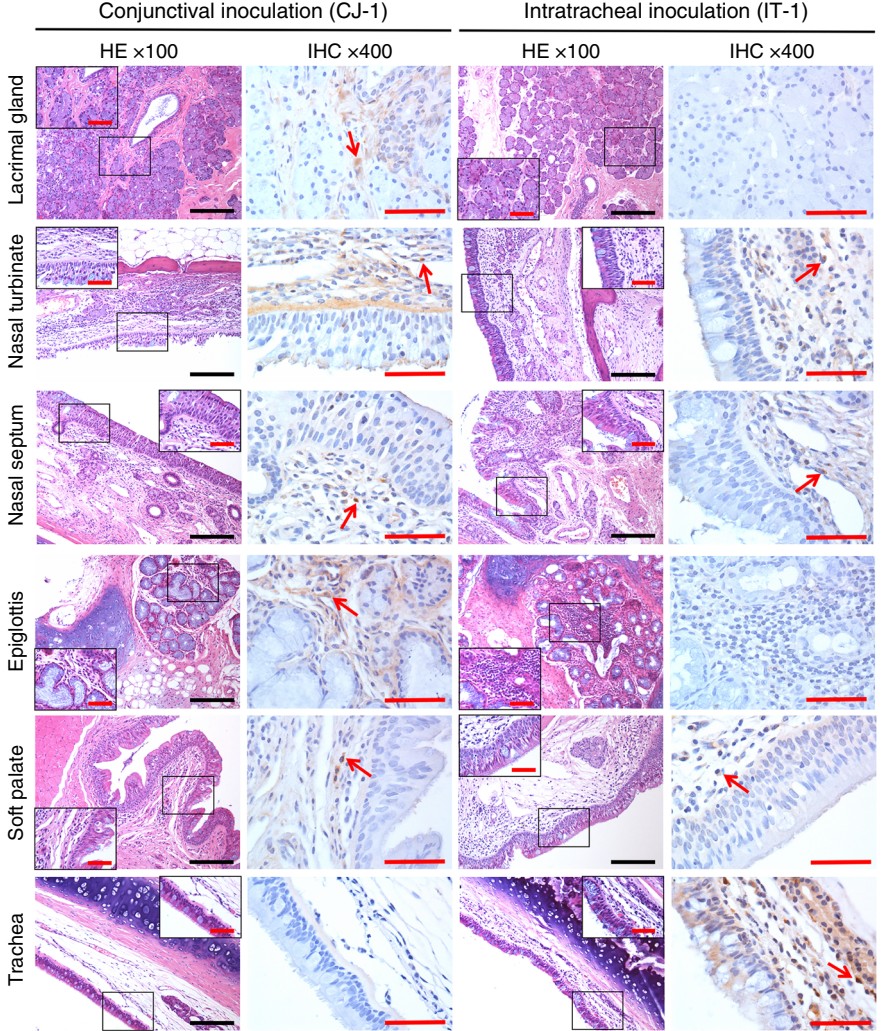

**Fig. 4 The distinct viral distributions in CJ-1 and IT-1 were consistent with anatomical structure.** Organs and tissues from the nasolacrimal system and conductive system of the respiratory tract, including the lacrimal gland, nasal turbinate, nasal septum, epiglottis, soft palate, and trachea, were examined. Sequential sections were stained with H&E and subjected to IHC. Fields in the H&E images outlined in black are magnified. The IHC images show the same field outlined in black at ×400 magnification. Black scale bar = 100 μm, red scale bar = 50 μm. Data are representative of three independent experiments.

stained with H&E. The histopathological changes in different tissues were observed under an Olympus microscope.

**Immunohistochemistry**. A ten percent buffered formalin solution was used to fix all the collected organs, and paraffin sections (3–4 μm in thickness) were prepared routinely as described in a previous report. Briefly, reagent from an antigen retrieval kit (Boster, AR0022) was applied to the sections at 37 °C for 1 min. A three percent solution of $H_2O_2$ in methanol was used to quench endogenous peroxidases for 10 min. The slices were incubated at 4 °C overnight with a laboratory-prepared 7D2 monoclonal antibody (1:200) after blocking in 1% normal goat serum. Horseradish peroxidase (HRP)-labelled goat anti-mouse IgG secondary antibody (Beijing ZSGB Biotechnology, ZDR-5307, 1:200) was added and maintained at 37 °C for 60 min. The slices were treated with 3,3′-diaminobenzidine tetra-hydrochloride for visualization. The sections were counterstained with haematoxylin, dehydrated, mounted on a slide, and observed with an Olympus microscope. The sequential sections from all collected tissues were directly incubated with HRP-labelled goat anti-mouse IgG and used as the omission control for viral antigen staining. The sequential sections from all collected tissues were incubated with a recombinant anti-mouse IgG antibody [RM104] (Abcam, ab190475, 1:1000) as the negative control for the expression of viral antigen.

**Statistical analysis**. The collected data were analysed with GraphPad Prism 8.0 software (GraphPad Software Inc., San Diego, CA).

**Reporting summary**. Further information on research design is available in the Nature Research Reporting Summary linked to this article.

## Data availability

The complete genome for this SARS-CoV-2 was submitted to NCBI (SARS-CoV-2/WH-09/human/2020/CHN, accession code MT093631.2). All raw data are available from the corresponding author upon reasonable request. Source data are provided with this paper.

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

## Acknowledgements
This work was supported by the National Key R&D Program of China (Grant No. SQ2020YFA070013), National Key Research and Development Project of China (Grant No. 2016YFD0500304), CAMS Initiative for Innovative Medicine of China (Grant No. 2016-I2M-2-006), and National Mega Projects of China for Major Infectious Diseases (Grant No. 2017ZX10304402).

## Author contributions
Conceptualization, resources and supervision: C.Q.; methodology: C.Q., W.D., L.B., H.G., Z.X., Y.Q., and Z.S.; investigation: W.D., L.B., H.G., Z.X., Y.Q., Z.S., S.G., J.L., J.L., P.Y., F.Q., Y.X., F.L., C.X., Q.L., J.X., Q.W., M.L., G.W., S.W., H.Y., T.C., X.L., W.Z., Y.H., and C.Q.; writing—original draft: Z.S. and J.X.; writing—review and editing: C.Q.; and funding acquisition: C.Q. and L.B.

## Competing interests
The authors declare no competing interests.
