## [Peer Review File · Nature Communications]

Reviewers' comments, first round -

Reviewer #1 (Remarks to the Author):

Here, Deng et al. provide evidence that conjunctival inoculation of SARS-CoV-2 results in productive infection of rhesus macaques, while intragastric inoculation did not. Animals inoculated via the conjunctivae shed vRNA in nasal swabs consistent with virus replication. One conjunctivally inoculated animal had histopathological lesions consistent with mild pneumonia at day 7 post-inoculation, while another produced detectable antibodies against the viral Spike protein by day 21. Patterns of viral antigen and RNA detection in tissues at day 7 were consistent with drainage of virus from the ocular and nasopharyngeal space into the respiratory tract in the conjunctivally inoculated animal. Lung lesions were more diffuse and severe in an animal inoculated intratracheally, while infected cells were not found in the ocular or nasopharyngeal tissues in that animal.

The authors conclude that conjunctival, but not intragastric, exposure to SARS-CoV-2 can lead to productive infection and recommend that eye protection be used by healthcare workers and others with potential ocular exposures to infected secretions.

These findings are timely and provide some important, if somewhat preliminary, insights into routes of SARS-CoV-2 exposure, but the small number of animals studied and in particular, the lack of testing for infectious virus, reduce the rigor of this study and its conclusions. The manuscript is also unclear on several important details and should be subjected to careful editing.

Major Points

1. The small number of animals in this study is a weakness that reduces the rigor of its conclusions. The lack of additional intratracheally inoculated animals is a particular weakness, since there is no comparison with i.t. inoculations after day 7. For example, it would be useful to know whether antibody titers differ meaningfully in conjunctivally vs. intratracheally inoculated animals. Histological analyses of pathological changes and presence of virus would also be strengthened by comparing more than one animal per inoculation route.
2. Although understanding the distribution of viral RNA in swabs and tissues is important for understanding distribution in the host, since this manuscript focuses on potential routes of transmission, it would be informative to include analyses of infectious virus in tissues in addition to vRNA loads.
3. Both writing and data displays lack clarity at important points at which this hinders the reader's ability to assess and understand the data.
 - a. Lines 58-9: The intratracheally inoculated animal appears to lose 125g weight at a single timepoint and recover 2 days later. Weights might be expected to fluctuate, particularly in animals undergoing frequent sedation. It is therefore difficult to conclude that this weight loss is biologically significant or specifically associated with infection. This suggestion should be removed from the main text and abstract.
 - b. Tissue viral load data are listed in the text (lines 76-95) and presented as a heat map in Figure 2. Both of these are somewhat difficult to read and interpret. Suggest presenting full viral load data as a table.
 - c. It is not clear whether chest radiography was performed between days 0 and 7. Earlier post-inoculation timepoints may show more severe findings, and a time-course would help contextualize findings on day 7.
 - d. Histopathology & IHC (lines 111-15): there is a description of the pathology observed in CJ-1. This section would benefit from a comparison to the pathology observed in IT-1. The description of Figure 3 also alludes to scale bars on line 332 that are not present in the figure.

Minor Points

1. The manuscript should be copy edited to address syntax, spelling, and grammar. Of particular note, the text refers several times to "IT inoculated macaques," suggesting that more than one animal was inoculated intratracheally (see, e.g., line 59).
2. Lines 65-9: it is not clear whether the authors are suggesting that vRNA detected in the conjunctivae 1 day after inoculation at that site is likely due to residual inoculum. This indeed seems likely, but it is not clear that this is the intended meaning.
3. The authors should report the inoculum dose in vRNA copy number in addition to infectious units so that the question of residual inoculum can be better interpreted. In addition, they should state whether the virus stock was sequenced to determine whether the consensus matched the original isolate and whether internal deletions in the spike gene were present at detectable frequencies.
4. In figure 2E, there appears to be data for IG-1 on days 14 and 21, despite the fact that this animal was euthanized on day 7. Furthermore, CJ-1 and IT-1 should be listed as "not collected" on those dates, rather than "not detected."
5. In the methods section, under RNA extraction and RT-PCR, the "previous report" mentioned on line 188 should be cited.

Reviewer #2 (Remarks to the Author):

This manuscript describes a small study in which the authors infected 5 Rhesus macaques with the SARS-CoV-2 virus. They used 2 monkeys for the intraconjunctival route of inoculation, 2 for the intragastric route, and 1 for the intratracheal route. A variety of parameters were evaluated, including temperature over 7 days, lung radiographs, virus shedding at various sites, and lung immunohistochemistry to show virus location.

The most interesting finding was the fact that the 2 monkeys that were infected by the ocular route did become infected, in contrast to the lack of apparent infection in the intragastric inoculated animals.

There are several problems with this ms. Firstly, the n is much too small. It is understandable that this may be necessary when working with monkeys, but the day 7 necropsy was only 1 animal per group and it is not effective to conclude anything from one animal. This work is excellent preliminary data for a larger study. But there is no validity to statistical analysis here.

Reviewer #3 (Remarks to the Author):

This manuscript investigates whether different inoculation routes, including intraconjunctival, intragastric and intratracheal, result in infection in Rhesus macaques. There are only two animals per route (conjunctival/gastric) and one animal (IT); however, the information provided from this study appears to be sufficient to demonstrate that conjunctival inoculation results in comparable virus replication as IT, while intragastric inoculation does not appear to result in a productive infection. This is valuable data and has very important implications for public health, and answers the questions as to whether the eyes can serve as an efficient route of infection.

There are only a few minor comments.

The cycling times (ln192) for PCR seem incorrect. 40 cycles at 95C for 15min (this is presumably the hot start).

Weight loss would probably be better shown as percent from starting rather than weight change in grams (Figure 2A).

It is not clear what the scale is in Fig 2D (at the top). Log viral RNA copies? Would it not be easier to just have a graph with the actual values?

REVIEWER COMMENTS

Reviewer #1 (Remarks to the Author):

Here, Deng et al. provide evidence that conjunctival inoculation of SARS-CoV-2 results in productive infection of rhesus macaques, while intragastric inoculation did not. Animals inoculated via the conjunctivae shed vRNA in nasal swabs consistent with virus replication. One conjunctivally inoculated animal had histopathological lesions consistent with mild pneumonia at day 7 post-inoculation, while another produced detectable antibodies against the viral Spike protein by day 21. Patterns of viral antigen and RNA detection in tissues at day 7 were consistent with drainage of virus from the ocular and nasopharyngeal space into the respiratory tract in the conjunctivally inoculated animal. Lung lesions were more diffuse and severe in an animal inoculated intratracheally, while infected cells were not found in the ocular or nasopharyngeal tissues in that animal.

The authors conclude that conjunctival, but not intragastric, exposure to SARS-CoV-2 can lead to productive infection and recommend that eye protection be used by healthcare workers and others with potential ocular exposures to infected secretions.

These findings are timely and provide some important, if somewhat preliminary, insights into routes of SARS-CoV-2 exposure, but the small number of animals studied and in particular, the lack of testing for infectious virus, reduce the rigor of this study and its conclusions. The manuscript is also unclear on several important details and should be subjected to careful editing.

Major Points

1. The small number of animals in this study is a weakness that reduces the rigor of its

conclusions. The lack of additional intratracheally inoculated animals is a particular weakness, since there is no comparison with i.t. inoculations after day 7. For example, it would be useful to know whether antibody titers differ meaningfully in conjunctivally vs. intratracheally inoculated animals. Histological analyses of pathological changes and presence of virus would also be strengthened by comparing more than one animal per inoculation route.

Author response: Thank you for your insightful comment. The objective of our research is to timely explore whether ocular conjunctival or intragastric route is the potential transmission way of SARS-CoV-2 rather than compare the pathogenesis and antibody titers between inoculated-animals via different routes. We hope to provide important experimental evidence to public prevention of COVID-19. Two macaques were inoculated via conjunctival route and all of them were confirmed to be infected by SARS-CoV-2. Our research provides a timely and essential suggestion for clinicians to protect their eyes when working with patients.

Additionally, we have performed abundance of relative experiments using rhesus macaques model of COVID-19 via intratracheal inoculation route, including exploring age-related rheseue macaque models of COVID-19 (Yu et al., 2020), evaluating vaccine candidate (Gao et al., 2020) and potential therapeutic drug (Deng et al., 2020) for SARS-CoV-2. From SARS (Qin et al., 2005) to SARS-CoV-2 (Bao et al., 2020), we have accumulated well experimental experiences to explore coronavirus using animal models.

Bao, L., Deng, W., Gao, H., et al. (2020), "Lack of Reinfection in Rhesus Macaques Infected with SARS-CoV-2", bioRxiv,.

Deng, W., Xu, Y., Kong, Q., et al. (2020), "Therapeutic efficacy of Pudilan Xiaoyan Oral Liquid

(PDL) for COVID-19 in vitro and in vivo", *Signal Transduct Target Ther*, Vol. 5 No. 1, pp. 66.

Gao, Q., Bao, L., Mao, H. et al. (2020), "Rapid development of an inactivated vaccine candidate for SARS-CoV-2", *Science*,.

Qin, C., Wang, J., Wei, Q., et al. (2005), "An animal model of SARS produced by infection of *Macaca mulatta* with SARS coronavirus", *J Pathol*, Vol. 206 No. 3, pp. 251-9.

Yu, P., Qi, F., Xu, Y., et al. (2020), "Age - related rhesus macaque models of COVID - 19", *Animal Models and Experimental Medicine*,.

2. Although understanding the distribution of viral RNA in swabs and tissues is important for understanding distribution in the host, since this manuscript focuses on potential routes of transmission, it would be informative to include analyses of infectious virus in tissues in addition to vRNA loads.

Author response: Thank you for your valuable comment. We adopted your excellent suggestion. During the research, we have analyzed the viral titer of different lobes of lungs from CJ-1 and IT-1. We added the relative data in Figure 2E and description in line 86-93 of the revised manuscript as follows:

Furthermore, to evaluate the infectious virus titer in the lung on 7 dpi, collected different lobes of lungs from CJ-1 and IT-1 were inoculated onto Vero E6 cells for virus isolation. In CJ-1, SARS-CoV-2 was isolated only from the left lower lobe of the lung (2.67 log₁₀ TCID₅₀/mL). In IT-1, the virus was isolated from more lobes of the lung (2.00 to 5.67 log₁₀ TCID₅₀/mL), most of which showed higher virus titers. They included the left lower lobe, the right lower lobe, the left middle lobe, the right middle lobe, the right accessory lobe, the left upper lobe, and the right upper lobe of

the lung (Figure 2E).

The associated methods were described in line 229-240 as follows:

To evaluate the infectious virus titer, different lobes of the lung from CJ-1 and IT-1 homogenates were prepared for virus titration analysis by endpoint titration in Vero E6 cells. Virus titer of the supernatant were determined using a standard 50% tissue culture infection dose (TCID₅₀) assay (Bao et al., 2020).

TCID₅₀ assay

The TCID₅₀ assay were performed as previous report (Gong et al., 2019). Briefly, to measure the titers of SARS-CoV-2, 10-fold serial dilutions of the viruses were used to inoculate Vero cell monolayers in DMEM containing 2% FBS at 37° C for 4 days. And then, the cytopathic effect (CPE) was observed, and the TCID₅₀ values were calculated by the Reed and Muench method (Reed and Muench, 1938).

Bao, L., Deng, W., Huang, B., et al (2020), "The pathogenicity of SARS-CoV-2 in hACE2 transgenic mice", *Nature*,.

Gong, S., Qi, F., Li, F., et al (2019), "Human-Derived A/Guangdong/Th005/2017 (H7N9) Exhibits Extremely High Replication in the Lungs of Ferrets and Is Highly Pathogenic in Chickens", *Viruses*, Vol. 11 No. 6, pp. 494.

Reed, L. J. & Muench, H. (1938), "A simple method of estimating fifty per cent endpoints", *American journal of epidemiology*, Vol. 27 No. 3, pp. 493-497.

3. *Both writing and data displays lack clarity at important points at which this hinders the reader's ability to assess and understand the data.*

a. *Lines 58-9: The intratracheally inoculated animal appears to lose 125g weight at a single timepoint and recover 2 days later. Weights might be expected to fluctuate, particularly in animals undergoing frequent sedation. It is therefore difficult to conclude that this weight loss is biologically significant or specifically associated with infection. This suggestion should be removed from the main text and abstract.*

Author response: Thank you for your valuable comment. We have fully considered your advice and deleted these sentences in line 54-56 and in abstract in line 26 to make the description more accurate.

b. *Tissue viral load data are listed in the text (lines 76-95) and presented as a heat map in Figure 2. Both of these are somewhat difficult to read and interpret. Suggest presenting full viral load data as a table.*

Author response: Thank you for your insightful comment. We have fully accepted your advice and concisely described the tissue viral load data in the revised manuscript in line 70-84. The viral load data of each tissue was shown on the heat map in the updated Figure 2D to make it clear.

c. *It is not clear whether chest radiography was performed between days 0 and 7. Earlier post-inoculation timepoints may show more severe findings, and a time-course would help contextualize findings on day 7.*

Author response: Thank you for your insightful comment. We have performed the chest radiography during this research between 0 dpi and 7 dpi as described in Figure 1 “**Graphic outline of experimental design and sample collection.**” We adopted your excellent suggestion and provided the data of chest radiography on 3 and 5 dpi. The relative data was updated in Figure 3A and shown in line 103-117 as follows:

Meanwhile, to observe the progressive pulmonary infiltration of SARS-CoV-2 related pneumonia, chest radiographs of inoculated animals were recorded every other day post-inoculation. From the beginning of 3 dpi, various degrees of abnormalities appeared in the lungs. Concretely, for the conjunctival inoculated-animal, compare with that before infection (day 0), the left upper lobe of the lung was suspected to present opaque glass sign on 3 dpi, and then developed to bilateral upper lobes on 5 dpi. Obscure lung markings and opaque glass sign in the bilateral lobes of the lung were observed on 7 dpi. By comparison, the intratracheal inoculated-animal developed relatively severe progressive-pulmonary-infiltration during 3-7 dpi. The right upper lobe of the lung exhibited an increase in density and obscure on 3dpi. The right lower lobe of the lung presented obscure lung markings and lamellar ground-glass opacities on 5 dpi. Increased radiographic changes were observed on 7 dpi, displaying patchy lesions in the right upper lobe of the lung, obscure lung markings, marked ground-glass opacities with a blurred right diaphragm in the bilateral lobes of the lung (Figure 3A).

d. Histopathology & IHC (lines 111-15): there is a description of the pathology observed in CJ-1. This section would benefit from a comparison to the pathology observed in IT-1. The description of Figure 3 also alludes to scale bars on line 332 that are not present in the figure.

Author response: Thank you for your valuable comment. We fully considered your advice and added the relative description in line 121-123 as follows:

IT-1 developed moderate and diffuse interstitial pneumonia featured by more widely thickened alveolar interstitium, more serious inflammation and exudation (Figure 3B).

Minor Points

1. The manuscript should be copy edited to address syntax, spelling, and grammar. Of particular note, the text refers several times to “IT inoculated macaques,” suggesting that more than one animal was inoculated intratracheally (see, e.g., line 59).

Author response: Thank you for your valuable comment. We have corrected the mistakes and improve English usage throughout the manuscript.

2. Lines 65-9: it is not clear whether the authors are suggesting that vRNA detected in the conjunctivae 1 day after inoculation at that site is likely due to residual inoculum. This indeed seems likely, but it is not clear that this is the intended meaning.

Author response: Thank you for your valuable comment. We have fully accepted your suggestion and deleted this sentence in line 63-64 and relative description in the abstract in line 25 to make it clear.

3. The authors should report the inoculum dose in vRNA copy number in addition to infectious units so that the question of residual inoculum can be better interpreted. In addition, they should state whether the virus stock was sequenced to determine whether the consensus matched the original isolate and whether internal deletions in the spike gene were present at detectable frequencies.

Author response: Thank you for your valuable comment.

TCID₅₀ is a regular and typical unit to describe the infectious dose in many published papers (Kim et al., 2020; Sia et al., 2020). The aim in this part is to inoculate the macaques via conjunctival route and analyze whether ocular conjunctiva is a potential portal for the transmission of SARS-CoV-2. For determining if this virus can enter the inoculated animals via ocular conjunctiva, the curve of viral replication post-inoculation and the distribution of viral loads in the inoculated animals were detected and analyzed. Actually, as the site of inoculation, there should exist residual inoculum in the conjunctiva at the beginning, therefore, we deleted the sentence you mentioned in line 64-65 and relative sentence in the abstract in line 25 to make the description clear.

Additionally, The SARS-CoV-2 (SARS-CoV-2/WH-09/human/2020/CHN/MT093631.2) has been isolated and sequenced by the Institute of Laboratory Animal Science, Peking Union Medical College. All the experimental animals were inoculated with the same batch of virus and there was no mutation in the gene sequence.

Kim, Y., Kim, S., Kim, S., Kim, E., Park, S., Yu, K., Chang, J., Kim, E. J., Lee, S. & Casel, M. A. B. (2020), "Infection and rapid transmission of SARS-CoV-2 in ferrets", *Cell host & microbe*,.

Sia, S. F., Yan, L., Chin, A. W., Fung, K., Choy, K., Wong, A. Y., Kaewpreedee, P., Perera, R. A., Poon, L. L. & Nicholls, J. M. (2020), "Pathogenesis and transmission of SARS-CoV-2 in golden hamsters", *Nature*, 1-7.

4. In figure 2E, there appears to be data for IG-1 on days 14 and 21, despite the fact

that this animal was euthanized on day 7. Furthermore, CJ-1 and IT-1 should be listed as “not collected” on those dates, rather than “not detected.”

Author response: Thank you for your valuable comment. We apologized for our mistake. We have corrected these data in Figure 2E. Additionally, “ND, not detected” was replaced by “NC, not collected” to make accurate expression.

5. In the methods section, under RNA extraction and RT-PCR, the “previous report” mentioned on line 188 should be cited.

Author response: Thank you for your insightful comment. We have cited our previous paper in line 187, 190, and 194 to make the relative methods clear and traceable.

Reviewer #2 (Remarks to the Author):

This manuscript describes a small study in which the authors infected 5 Rhesus macaques with the SARS-CoV-2 virus. They used 2 monkeys for the intraconjunctival route of inoculation, 2 for the intragastric route, and 1 for the intratracheal route. A variety of parameters were evaluated, including temperature over 7 days, lung radiographs, virus shedding at various sites, and lung immunohistochemistry to show virus location.

The most interesting finding was the fact that the 2 monkeys that were infected by the ocular route did become infected, in contrast to the lack of apparent infection in the intragastric inoculated animals.

There are several problems with this ms. Firstly, the n is much too small. It is understandable that this may be necessary when working with monkeys, but the day 7 necropsy was only 1 animal per group and it is not effective to conclude anything from one animal. This work is excellent preliminary data for a larger study. But there is no validity to statistical analysis here.

Author response: Thank you for your insightful comment. The objective of our research is to timely explore whether ocular conjunctival or intragastric route is the potential transmission way of SARS-CoV-2. We hope to provide important experimental evidence to public prevention of COVID-19. Two macaques were inoculated via conjunctival route and all of them were confirmed to be infected by SARS-CoV-2. Our research provides a timely and essential suggestion for clinicians to protect their eyes when working with patients.

During the research, we have analyzed the viral titer of different lobes of the lung from CJ-1 and IT-1. We added the relative data in Figure 2E and description in line 86-93 of the revised manuscript as follows:

Furthermore, to evaluate the infectious virus titer in the lung on 7 dpi, collected different lobes of lungs from CJ-1 and IT-1 were inoculated onto Vero E6 cells for virus isolation. In CJ-1, SARS-CoV-2 was isolated only from the left lower lobe of the lung ($2.67 \log_{10} \text{TCID}_{50}/\text{mL}$). In IT-1, the virus was isolated from more lobes of the lung (2.00 to $5.67 \log_{10} \text{TCID}_{50}/\text{mL}$), most of which showed higher virus titers. They included the left lower lobe, the right lower lobe, the left middle lobe, the right middle lobe, the right accessory lobe, the left upper lobe, and the right upper lobe of the lung (Figure 2E).

Furthermore, we added and provided the data of chest radiography on 3 and 5 dpi from CJ-1 and IT-1. The relative data was updated in Figure 3A and shown in line 103-117 as follows:

Meanwhile, to observe the progressive pulmonary infiltration of SARS-CoV-2 related pneumonia, chest radiographs of inoculated animals were recorded every other day post-inoculation. From the beginning of 3 dpi, various degrees of abnormalities appeared in the lungs. Concretely, for the conjunctival inoculated-animal, compare with that before infection (day 0), the left upper lobe of the lung was suspected to present opaque glass sign on 3 dpi, and then developed to bilateral upper lobes on 5 dpi. Obscure lung markings and opaque glass sign in the bilateral lobes of the lung were observed on 7 dpi. By comparison, the intratracheal inoculated-animal developed relatively severe progressive-pulmonary-infiltration during 3-7 dpi. The

right upper lobe of the lung exhibited an increase in density and obscure on 3dpi. The right lower lobe of the lung presented obscure lung markings and lamellar ground-glass opacities on 5 dpi. Increased radiographic changes were observed on 7 dpi, displaying patchy lesions in the right upper lobe of the lung, obscure lung markings, marked ground-glass opacities with a blurred right diaphragm in the bilateral lobes of the lung (Figure 3A).

Additionally, we have performed abundance of relative experiments using rhesus macaques model of COVID-19 via intratracheal inoculation route, including exploring age-related rhseue macaque models of COVID-19 (Yu et al., 2020), evaluating vaccine candidate (Gao et al., 2020) and potential therapeutic drug (Deng et al., 2020) for SARS-CoV-2. From SARS (Qin et al., 2005) to SARS-CoV-2 (Bao et al., 2020), we have accumulated well experimental experiences to explore coronavirus using animal models.

Bao, L., Deng, W., Gao, H., et al. (2020), "Lack of Reinfection in Rhesus Macaques Infected with SARS-CoV-2", bioRxiv,.

Deng, W., Xu, Y., Kong, Q., et al. (2020), "Therapeutic efficacy of Pudilan Xiaoyan Oral Liquid (PDL) for COVID-19 in vitro and in vivo", Signal Transduct Target Ther, Vol. 5 No. 1, pp. 66.

Gao, Q., Bao, L., Mao, H. et al. (2020), "Rapid development of an inactivated vaccine candidate for SARS-CoV-2", Science,.

Qin, C., Wang, J., Wei, Q., et al. (2005), "An animal model of SARS produced by infection of Macaca mulatta with SARS coronavirus", J Pathol, Vol. 206 No. 3, pp. 251-9.

Yu, P., Qi, F., Xu, Y., et al. (2020), "Age - related rhesus macaque models of COVID - 19", Animal Models and Experimental Medicine,.

Reviewer #3 (Remarks to the Author):

This manuscript investigates whether different inoculation routes, including intraconjunctival, intragastric and intratracheal, result in infection in Rhesus macaques. There are only two animals per route (conjunctival/gastric) and one animal (IT); however, the information provided from this study appears to be sufficient to demonstrate that conjunctival inoculation results in comparable virus replication as IT, while intragastric inoculation does not appear to result in a productive infection. This is valuable data and has very important implications for public health, and answers the questions as to whether the eyes can serve as an efficient route of infection.

There are only a few minor comments.

1. The cycling times (ln192) for PCR seem incorrect. 40 cycles at 95C for 15min (this is presumably the hot start).

Author response: Thank you for your valuable comment. We apologized for our mistake. We have corrected the description of PCR in line 198-199. RT-PCR reactions were applied to the PowerUp SYBG Green Master Mix Kit from Applied Biosystems, USA, following cycling protocol: 50°C for 2 min, 95°C for 2 min, followed by 40 cycles at 95°C for 15 s and 60°C for 30 s, and then 95°C for 15 s, 60°C for 1 min, 95°C for 45 s.

2. Weight loss would probably be better shown as percent from starting rather than weight change in grams (Figure 2A).

Author response: Thank you for your insightful comment. There was no significant change of body weight between inoculated animals, therefore, we selected weight change in grams to display the slightly difference. During this experiment, it was difficult to conclude that this weight loss is biologically significant or specifically associated with infection. We deleted these sentences about obviously weight loss in intratracheal inoculated animals in line 54-56 and in abstract to in line 26 to make the description more accurate.

3. It is not clear what the scale is in Fig 2D (at the top). Log viral RNA copies? Would it not be easier to just have a graph with the actual values?

Author response: Thank you for your valuable comment. We have added the scale in the updated Figure 2D. The description of the tissue viral load data was revised in line 70-84 to make it concise and clear. The viral load data of each tissue was shown on the heat map in the updated Figure 2D to make it clear.

Reviewers' comments, second round -

Reviewer #1 (Remarks to the Author):

Overview

In this revised manuscript, Deng et al have responded to many concerns raised in the first round of review. However, some weaknesses remain. The most important weakness identified by 2 reviewers was the limited number of animals evaluated in the present study. This is still not adequately addressed in the current manuscript.

Major Points

1. A prior critique of the manuscript was the small sample size. The authors state that they have performed intratracheal (i.t.) inoculations in previous studies. These previous data should be cited here and compared with viral load and pathology results from the single i.t.-inoculated macaque from the present study.
2. The authors emphasize in their rebuttal that their focus here is on reporting that productive infection by the conjunctival route is possible. This is an important finding in itself, but the potential for conjunctival transmission is not rigorously defined by only challenging 2 animals with a single high dose and performing histopathological analysis on a single animal. Therefore the data presented here do not help us evaluate how great a risk ocular exposure poses relative to respiratory exposure. I understand that macaque experiments are difficult and expensive, but if the authors cannot perform additional conjunctival challenges and evaluate additional doses, then they should clearly state the limitations of this study, particularly noting that they cannot currently evaluate the risk of ocular exposure relative to other routes.

Minor Points

1. In both the abstract and the end of the discussion, the authors state that "Both the two routes affected the alimentary canal." Presumably this means i.t. and conjunctival inoculations, but this is not fully clear. More importantly, it is not clear what "affected" means here. The authors detect viral antigen and viral RNA in tissues categorized as "alimentary tract", but evidence of viral pathology in these tissues is not presented.
2. In the abstract on lines 24-26, the description of which swabs from which cohorts were positive is unclear.
3. In methods the authors should clearly state that the stock virus consensus sequence matched the expected reference.

REVIEWER COMMENTS

Reviewer #1 (Remarks to the Author):

Overview

In this revised manuscript, Deng et al have responded to many concerns raised in the first round of review. However, some weaknesses remain. The most important weakness identified by 2 reviewers was the limited number of animals evaluated in the present study. This is still not adequately addressed in the current manuscript.

Major Points

1. A prior critique of the manuscript was the small sample size. The authors state that they have performed intratracheal (i.t.) inoculations in previous studies. These previous data should be cited here and compared with viral load and pathology results from the single i.t.-inoculated macaque from the present study.

Author response: Thank you for your insightful comment. We adopted your excellent suggestion.

In our previous studies, rhesus macaques were intratracheally infected with SARS-CoV-2. After euthanized and necropsied at 7 dpi, more than 4.0 log₁₀ RNA copies/mL from the lung was detected. Microscopically, lesions were mainly in the lungs where present typically moderate interstitial pneumonia. The viral load and pathological changes in lung of the IT infected macaque in this study was in the range and consistent with the previous study (Yu *et al.*, 2020; Gao *et al.*, 2020; Cao *et al.*, 2020).

We have added the relative description in the section of discussion in line 167-173.

References:

Cao, Y., Su, B., Guo, X., Sun, W., Deng, Y., Bao, L., Zhu, Q., Zhang, X., Zheng, Y. & Geng, C. (2020), "Potent neutralizing antibodies against SARS-CoV-2 identified by high-throughput single-cell sequencing of convalescent patients' B cells", *Cell*,.

Gao, Q., Bao, L., Mao, H., Wang, L., Xu, K., Yang, M., Li, Y., Zhu, L., Wang, N. & Lv, Z. (2020), "Rapid development of an inactivated vaccine candidate for SARS-CoV-2", *Science*,.

Yu, P., Qi, F., Xu, Y., Li, F., Liu, P., Liu, J., Bao, L., Deng, W., Gao, H. & Xiang, Z. (2020), "Age - related rhesus macaque models of COVID - 19", *Animal Models and Experimental Medicine*,.

2. The authors emphasize in their rebuttal that their focus here is on reporting that productive infection by the conjunctival route is possible. This is an important finding in itself, but the potential for conjunctival transmission is not rigorously defined by only challenging 2 animals with a single high dose and performing histopathological analysis on a single animal. Therefore the data presented here do not help us evaluate how great a risk ocular exposure poses relative to respiratory exposure. I understand that macaque experiments are difficult and expensive, but if the authors cannot perform additional conjunctival challenges and evaluate additional doses, then they should clearly state the limitations of this study, particularly noting that they cannot currently evaluate the risk of ocular exposure relative to other routes.

Author response: Thank you for your insightful comment. We have fully accepted your excellent suggestion. Although both of the CJ inoculated animals were infected by SARS-CoV-2. The small sample size is the limitation of our study. We have tone down the strength of the conclusions relating to ocular transmission and added the description of the limitation as you mentioned in the section of the discussion.

In the revised main text, we added “However, further studies with more animals and different viral doses are needed to evaluate the risk of ocular exposure relative to other routes” (in line 160-161).

Minor Points

1. In both the abstract and the end of the discussion, the authors state that “Both the two routes affected the alimentary canal.” Presumably this means i.t. and conjunctival inoculations, but this is not fully clear. More importantly, it is not clear what “affected” means here. The authors detect viral antigen and viral RNA in tissues categorized as “alimentary tract”, but evidence of viral pathology in these tissues is not presented.

Author response: Thank you for your insightful comment. We have fully considered your valuable suggestion. We have unified this term as “alimentary tract” throughout this manuscript in line 82, 95, 131, and 225. As we have described this in the section of the result, we delete these description in the abstract and in the end of discussion to make it concise. In recent studies, viral antigen and pathology were also examined in patients with COVID-19, viruses related pathological changes were not always parallely observed in the tissues presenting viral RNA and viral antigen (Wang *et al.*, 2020). This phenomenon is needed to be further investigated.

Wang, C., Xie, J., Zhao, L., Fei, X., Zhang, H., Tan, Y., Zhou, L., Liu, Z., Ren, Y. & Yuan, L. (2020), "Aveolar macrophage activation and cytokine storm in the pathogenesis of severe COVID-19".

2. In the abstract on lines 24-26, the description of which swabs from which cohorts

were positive is unclear.

Author response: Thank you for your insightful comment. We have added relative description as you mentioned to make it clear in line 24-26 as follows, “CJ and IT inoculated animals were able to detect viral RNA in their nasal and throat swabs from 1 to 7 dpi. Viral RNA from the anal swab was only detected in the IT group at 1-7 dpi.”

3. In methods the authors should clearly state that the stock virus consensus sequence matched the expected reference.

Author response: Thank you for your insightful comment. We have corrected this in line 191-192.

1. REVIEWER COMMENTS

Dear Prof Qin,

Your manuscript entitled "Ocular conjunctival inoculation of SARS-CoV-2 can cause mild COVID-19 in Rhesus macaques" has now been seen again by our referees. In light of their advice I am delighted to say that we are happy, in principle, to publish a suitably revised version in Nature Communications under the open access CC BY license (Creative Commons Attribution v4.0 International License).

We therefore invite you to revise your paper one last time to address the remaining concerns of our reviewers. In line with previous reviewer's concerns and to make limitations of this study more clear, we suggest you to point out limitations in the beginning of Results (after "IT-1 was regarded as a comparison to compare the distributions and pathogenesis of viruses after enter the host via different routes (Figure 1). "). Please add that the low number of non-human primates and the single tested dose is a limitation of the study and that therefore the risk of ocular exposure in comparison to other transmission routes of infection can not be assessed.

Author response: Thank you for your insightful comment. We adopted your excellent suggestion. We have added this description in the section of result as you mentioned in line 71-74.